# Rat Animal Model of Pectus Excavatum

**DOI:** 10.3390/life10060096

**Published:** 2020-06-26

**Authors:** Vlad-Laurentiu David, Bogdan Ciornei, Florin-George Horhat, Elena Amaricai, Ioana-Delia Horhat, Teodora Hoinoiu, Eugen-Sorin Boia

**Affiliations:** 1Department of Pediatric Surgery and Orthopedics, “Victor Babes” University of Medicine and Pharmacy, Eftimie Murgu Sq. No 2, 300041 Timisoara, Romania; david.vlad@umft.ro (V.-L.D.); boiaeugen@yahoo.com (E.-S.B.); 2Department of Microbiology, “Victor Babes” University of Medicine and Pharmacy, Eftimie Murgu Sq. No 2, 300041 Timisoara, Romania; 3Department of Rehabilitation, Physical Medicine and Rheumatology, “Victor Babes” University of Medicine and Pharmacy Timisoara, 300041 Timisoara, Romania; amaricai.elena@umft.ro; 4Department of ENT, “Victor Babes” University of Medicine and Pharmacy, Eftimie Murgu Sq. No 2, 300041 Timisoara, Romania; 5Division of Clinical Practice Skills, “Victor Babes” University of Medicine and Pharmacy, Eftimie Murgu Sq. No 2, 300041 Timisoara, Romania; tstoichitoiu@umft.ro

**Keywords:** pectus excavatum, animal model, rat, chest wall deformity

## Abstract

Background: pectus excavatum (PE) is the most common congenital deformity of the thoracic wall. Lately, significant achievements have been made in finding new, less invasive treatment methods for PE. However, most of the experimental work was carried out without the help of an animal model. In this report we describe a method to create an animal model for PE in Sprague-Dawley rats. Methods: We selected 15 Sprague-Dawley rat pups and divided them into two groups: 10 for the experimental group (EG) and 5 for the control group (CG). We surgically resected the last four pairs of costal cartilages in rats from the EG. The animals were assessed by CT-scan prior to surgery and weekly for four consecutive weeks. After four weeks, the animals were euthanized and the thoracic cage was dissected from the surrounding tissue. Results: On the first postoperative CT, seven days after surgery, we observed a marked depression of the lower sternum in all animals from the EG. This deformity was present at every CT-scan after surgery and at the post-euthanasia assessment. Conclusions: By decreasing the structural strength of the lower costal cartilages, we produced a PE animal model in Sprague-Dawley rats.

## 1. Introduction

Pectus excavatum (PE) is the most common congenital deformity of the thoracic wall [1]. The deformity consists in the posterior deviation of the sternum and the lower costal cartilages [1]. The disease was recognized in the 16th century and the first successful repair was performed by Sauerbruch in 1913 [1]. Corrective surgery remained the main therapeutic mean of PE, with more than 50 different surgical techniques performed in the past century [1,2]. Nowadays, excellent results are achieved with Nuss minimal invasive repair of pectus excavatum [2,3]. Great steps have been made in understanding the pathogenesis of this disease. It is a known fact that PE can produce functional impairment in the heart or lungs and has significant consequences over the normal psychological development of children [2,4]. PE is associated frequently with connective tissue diseases, like Marfan and Ehler Danlos syndrome, and most of the patients have a particular Marfanoid phenotype [1,2,4]. This fact is an indicator that the deformation of the chest wall is caused by a disturbance of the normal structure and function of costal cartilages. Several histologic studies revealed that not only the structure, but also the physical and chemical features of the costal cartilages are altered in PE patients versus normal patients [5,6,7,8,9,10,11,12]. The costal cartilages reveal a disturbed growth pattern and/or disturbed structural strength leading to the inward bending of the anterior chest wall [2,4,5]. However, the intimate mechanism leading from the cartilage disturbances to the chest deformity is still an unknown issue.

PE has also been observed in animals [13]. Reports in animals refer mainly to companion animals, like cats and dogs, and the way veterinarians managed these conditions [14,15,16]. PE deformity and idiopathic scoliosis was observed in mice, revealing mutation of the gene encoding the adhesion G protein-coupled receptor, 126/adhesion G protein-coupled receptor G6, in osteochondroprogenitor cells [17]. To date there are only a few experimental studies trying to establish an animal model for PE: Geisbe et al. [18] and Wang et al. [19] worked on rabbit model and Karner et al. [17] in mice. In this report we present an animal model for PE in Sprague-Dawley rats.

## 2. Materials and Methods 

### 2.1. The Animals

Prior to the experiment, appropriate approval was obtained from the Ethics Committee of the “Victor Babes” University of Medicine and Pharmacy Timisoara (No. 125 of the 18 December 2015). All the experiments comply with the Animal Research: Reporting in Vivo Experiments (ARRIVE) guidelines. In order to minimize the number of sacrifices, we used 15 Sprague-Dawley rat pups, 21 days old, which were randomly divided into two study groups: 10 rats for the experimental group (EG) and 5 for the control group (CG). 

### 2.2. Surgical Procedure

We performed surgical intervention on the EG rats as follows: with the animals on general anesthesia (inhalation, 5% Isofluran and O_2_ at 1 L/min, for induction in the anesthesia chamber, and after that, a facemask of 2% Isofluran and O_2_ at 1 L/min was delivered), the fur from the abdomen was extensively removed using hair-clippers (Favorita^®^, Aesculap, Tuttlingen, Germany) and sterilized with betadine (Aegis Pharma, Bassussarry, France) in order to perform the surgical procedure [20]. We placed the animals in the ventral position, immobilized the hind limbs in extension using elastic loops and performed a longitudinal midline incision on the anterior chest wall. We detached the pectoralis major muscle from its medial insertion and exposed the last costal cartilages bilaterally. The dissection of the muscle and adjacent tissue was kept to a minimum in order to limit local scaring effect. With the help of a bipolar forceps, we cauterized until we accomplished complete resection of the last 4 costal cartilages without opening the pleural cavity (Figure 1). Afterwards, we sutured the muscle and the skin on the midline. No surgical procedures were performed on the 5 rats from the CG. Following the interventions, we kept all rats in constant humidity and temperature with a cycle of 12 h light and darkness exposure, food and water ad libitum. We recorded the weight and length of all rats from both groups weekly. 

### 2.3. CT Scan, Assessment

Prior to surgery, all rats underwent a CT scan (CT1), then weekly for the following 4 weeks (CT2–5). At each CT scan we assessed the shape of the thoracic cage, and measured and recorded the following diameters: axially at the level of the spine (AD); sagittal at the level of the upper thoracic aperture (SSD); at the level of the 4th rib (SD4); at the level of the lower thoracic aperture (ISD); transverse diameters at the level of the upper thoracic opening (STD); at the level of the 4th rib (TD4); and at the level of the lower thoracic opening (ITD). We calculated the Haller index by dividing the transverse diameter to the antero-posterior diameter measured at the level of upper thoracic aperture (SHI), the 4th rib (HI4), and at the level of the lower thoracic opening (IHI). 

### 2.4. Direct Assessment and Measurements of the Thoracic Cage 

Four weeks after surgery (7 weeks of life), all rats were euthanized by CO_2_ inhalation, and weight and head-to-tail length were measured. We dissected the thoracic cage and removed all the soft tissues and intrathoracic organs. Similar with the measurements on the CT images, we measured and recorded the height of the thorax, the anteroposterior and transverse diameters at the level of the upper opening, at the level of the 4th rib and at the level of the lower opening for each subject. We calculated the Haller index at the level of upper thoracic aperture (HIS), the 4th rib (HI4), and at the level of the lower thoracic opening (HII). 

### 2.5. Statistical Analysis

Statistical analysis was performed using the unpaired *t*-test with a significance threshold set at *p* = 0.05 for 95% CI. We used the Pearson’s product-moment correlation to calculate if there was a correlation between the different parameters. 

## 3. Results

One rat from the EG died during surgery due to incidentally produced suffocating pneumothorax. The mean weight before surgery was 71.57 ± 5.8 g and increased to 208.28 ± 14.14 g at the end of the experiment (before euthanasia), without significant differences between the two groups (*p* > 0.05). The length of the subjects was 208.28 ± 14 mm before surgery, respectively 373.92 ± 9.2 mm at the end of the experiment (before euthanasia), also without differences between the two groups (*p* > 0.05).

### 3.1. Assessment on CT Images

On the CT-scan performed on the 7th day of the experiment (CT2), we observed a marked depression of the lower sternum in all rats from the EG (Figure 2). This deformation of the sternum remained present in the EG throughout the entire period of the experiment, in each CT examination (CT2–5).

There were no statistically significant differences between the two groups for AD all through the experiment CT1–5 (*p* > 0.05) (Table 1). SSD was similar from CT1 to CT4 (*p* > 0.05). At CT5, SSD was higher in the EG (*p* = 0.01). SD4 was similar between groups, except CT3 where it was higher for the CG (*p* = 0.02). Significant differences were noticed for the ISD measured on the CT2, CT3, CT4 and CT5, ISD for the EG was significantly smaller than in the CG (Figure 3). There were no significant statistical differences between the transverse diameters of the EG and the CG, except for STD at CT 2 (EG > CG, *p* = 0.01) and ITD at CT4 (CG > EG, *p* = 0.01).

The SHI were similar between the two groups at CT2, CT3 and CT4 (*p* > 0.05) (Table 2). At CT1 and CT5, SHI was higher in the CG (*p* = 0.03, respectively *p* = 0.02). There were no significant differences between groups for HI4 at any moment at CT1–5 (*p* > 0.05). IHI was similar between groups at CT1 (*p* > 0.05), but was significantly higher in the EG versus the CG starting with the second CT examination (CT2) (Figure 4).

### 3.2. Direct Assessment and Measurements of the Thoracic Cage 

After euthanasia of the EG rats, the deformity of the anterior chest wall was visible at inspection (Figure 5). The AD, STD, ITD and SSD were similar between the two groups (*p* > 0.05). The mean ISD was 40.60 ± 3.20 mm in the CG versus 36.56 ± 1.59 mm in the EG, which was statistically significant (*p* = 0.04).

## 4. Discussion

Animal models for human medical conditions are very useful research tools for scientists. They are used extensively in many fields of medical research and are the basis for most in vivo experiments [21]. Our model is based on the assumption that a weakened anterior chest wall will collapse due to the intrathoracic negative pressure and the traction of the diaphragm muscle during inspiration. During inspiration, the diaphragm contracts in order to enlarge thoracic capacity and thus exerts a traction force on its insertion at the level of the inner face of the chest. The greatest force is applied at the level of the anterior wall, on the median line, where the tendinous center of the muscle is located. We must consider that the costal cartilages are the structures with the lowest structural strength within the components which make up the thoracic cage. The other components: the thoracic spine, the ribs and the sternum, being bony structures, are rigid structures. Therefore, it seems logical that this is the weakest point of the thoracic cage, the point where a force that tends to deform the thorax has the highest chance of succeeding. This mechanism was proposed as an etiological factor for PE in human subjects by Brown [22], but he erroneously assumed that the essential etiological factor is the increase traction of the diaphragm. Later on, it was demonstrated that the diaphragm muscle is unremarkable in patients with PE versus normal [23].

In our study, the resection of the lower costal cartilage had the purpose to create a mechanical weakness in the lower area of the sternum. We found out that, by lowering the structural strength of the last four costal cartilages, we induced the posterior bending of the sternum in 100% of animals from the experimental group. This deformity was visible after the first seven days after surgery and persisted on CT images throughout the next four weeks until euthanasia of the experimental animals was performed. Subsequently, the deformity was observed and quantified on macroscopic evaluation after euthanasia and dissection of the thoracic cage (Figure 3). We demonstrated that, once the strength of the costal cartilages is reduced, the sternum left without the support of the last costal cartilages will move to the posterior and will remain fixed in this position. This is in accordance with the findings of Feng et al. [9]. They demonstrated that costal hyaline cartilage in patients with pectus excavatum is less resistant to tension and compression forces [9]. Geisbe et al. [22] reached the same conclusion; the deformity of the sternum is caused by a disproportion between the traction forces exerted by a normal diaphragm acting against a weak structural strength of anterior thoracic wall. Our study supports this theory; by reducing the structural strength of the anterior chest wall we induced a PE-like deformity in experimental animals. 

We have chosen to produce our animal model of PE in Sprague-Dawley rats because these animals are widely available, cheap, resistant to stress, have great anatomical and physiological similarities with humans, do not require special living and breathing conditions and are large enough to carry on experimental work [24]. We also made a deliberate decision to use Sprague-Dawley rat pups because our intention was to mimic PE as close as possible in all its aspects. PE pathogenesis is intimately linked with growth and development of children’s skeletal growth [2]. The deformation of the thoracic wall evolves only during childhood through adolescence [2]. In adults, after the skeletal growth stops, the deformation of the thoracic wall becomes stable [2]. We believe this is an important aspect in order to fully understand the etiopathogenesis or conduct experimental work on animal models for PE. In our animal model, the induced PE in three weeks old rat pups remained present during rats’ “childhood” and became stabile when the animals reached maturity. 

There are certain limitations of this study, the animal model for PE, and its final result. We produced the excavatum deformity of the chest wall by means of surgery under general anesthesia. This can have local or general side effects which are not present in natural occurring PE. Perhaps future experimental studies can produce this weakening of the costal cartilages by less invasive means. There are certain differences regarding the pathogenesis of PE in our model versus natural occurred PE. In our model there is a relative sudden drop of the structural strength of the costal cartilage and consecutive deformation of the chest wall, while in natural occurred PE this process is more gradual and the deformation is sometimes asymmetric. 

## 5. Conclusions

The decrease of the structural strength of the last four costal cartilages leads to a funnel like chest deformity of the sternum in experimental animals. We have created an animal model for pectus excavatum in Sprague-Dawley rats.

## Figures and Tables

**Figure 1 life-10-00096-f001:**
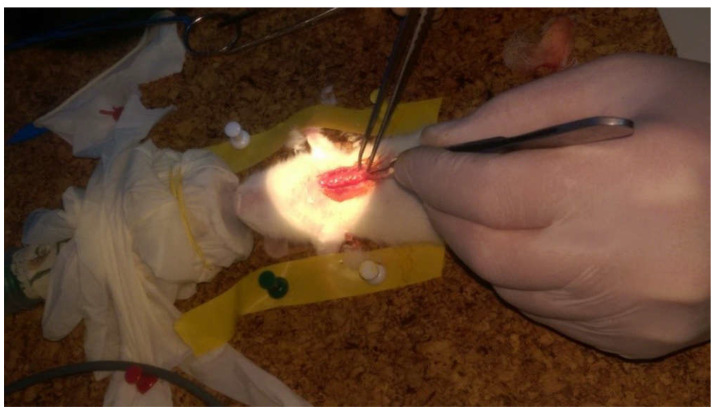
Sectioning the last 4 costal cartilages in Sprague-Dawley rats.

**Figure 2 life-10-00096-f002:**
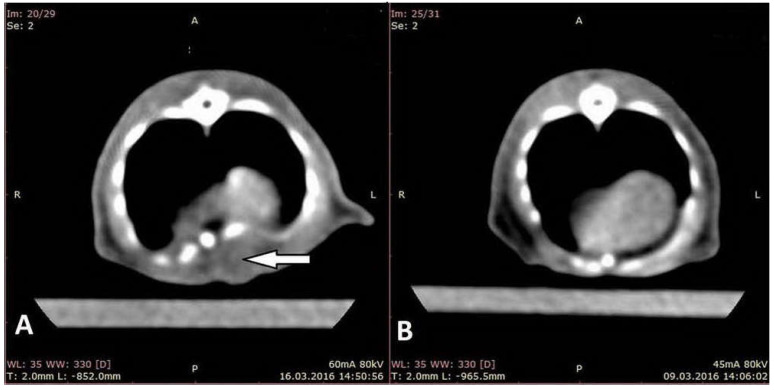
CT scan showing the depression of the sternum, 7 days from surgery (**A**) and CT scan of the thorax from the control group (**B**).

**Figure 3 life-10-00096-f003:**
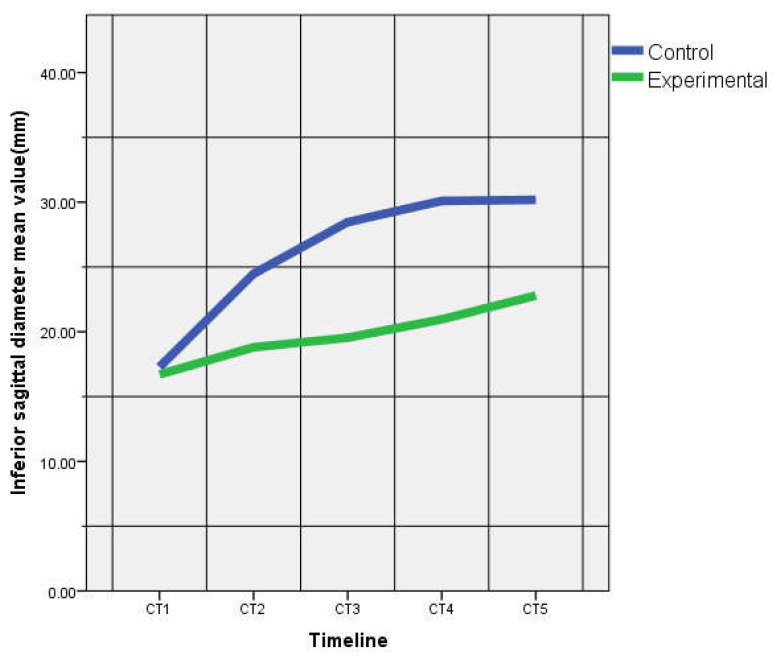
Mean inferior sagittal diameters on CT-scan images (CT1–CT5).

**Figure 4 life-10-00096-f004:**
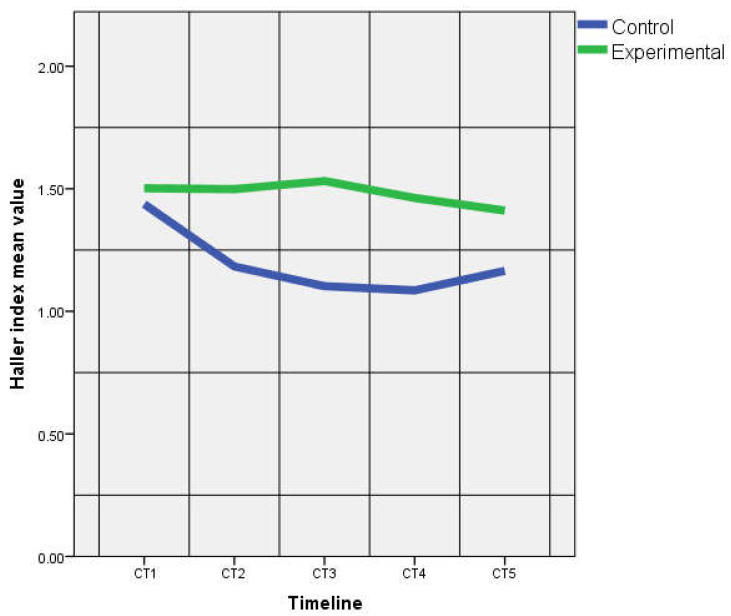
Haller index at the lower thoracic aperture (CT1–CT5).

**Figure 5 life-10-00096-f005:**
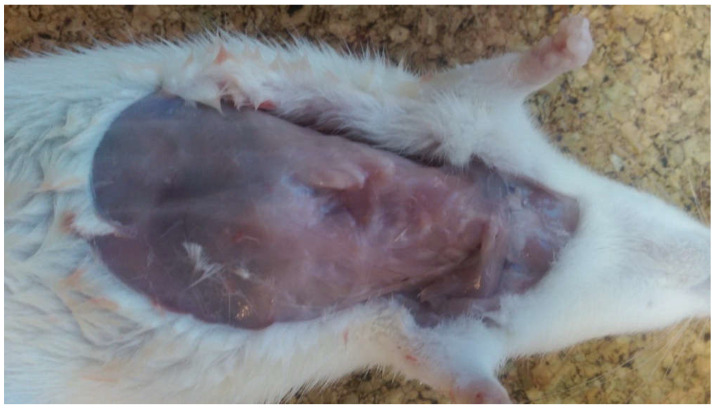
Sternal funnel chest like deformity 4 weeks from surgery.

**Table 1 life-10-00096-t001:** Diameters of the rib cage measured on the CT images preoperative (CT1) through 4 weeks after surgery.

	CT1 (Preoperative)		CT2 (7 Days)		CT3 (14 Days)		CT4 (21 Days)		CT5 (28 Days)	
	CG	EG	CG	EG	CG	EG	CG	EG	CG	EG
**AD** (mm)	29.12 ± 1.08	28.24 ± 2.03	31.76 ± 1.95	31.50 ± 1.79	35.62 ± 0.56	36.27 ± 1.00	38.80 ± 0.84	38.38 ± 1.24	40.56 ± 1.00	40.08 ± 1.54
	*p* = 0.39		*p* = 0.80		*p* = 0.20		*p* = 0.51		*p* = 0.55	
**SSD** (mm)	4.77 ± 0.14	4.95 ± 0.29	5.09 ± 0.24	5.24 ± 0.28	5.54 ± 0.18	5.65 ± 0.35	5.77 ± 0.27	6.12 ± 0.24	6.00 ± 0.22	6.71 ± 0.42
	*p* = 0.24		*p* = 0.33		*p* = 0.53		*p* = 0.35		*p* = 0.01	
**SD4** (mm)	8.63 ± 0.44	8.57 ± 0.48	10.61 ± 0.28	10.93 ± 0.90	12.04 ± 0.45	11.08 ± 0.78	12.72 ± 0.64	12.72 ± 0.92	13.92 ± 0.63	13.76 ± 0.65
	*p* = 0.81		*p* = 0.47		*p* = 0.02		*p* = 0.99		*p* = 0.67	
**ISD** (mm)	17.28 ± 0.58	16.71 ± 1.16	24.48 ± 0.74	18.80 ± 3.14	28.46 ± 2.54	19.54 ± 2.00	30.10 ± 0.45	20.95 ± 2.25	30.18 ± 0.57	22.78 ± 2.12
	*p* = 0.33		*p* = 0.02		*p* = 0.00		*p* = 0.00		*p* = 0.00	
**STD** (mm)	6.92 ± 0.12	6.84 ± 0.22	7.12 ± 0.09	7.52 ± 0.37	7.30 ± 0.21	7.76 ± 0.56	7.92 ± 0.14	8.29 ± 0.42	8.54 ± 0.37	8.45 ± 0.54
	*p* = 0.373		*p* = 0.03		*p* = 0.10		*p* = 0.08		*p* = 0.73	
**TD4** (mm)	15.12 ± 0.68	14.52 ± 0.90	17.56 ± 0.40	17.52 ± 0.99	19.96 ± 0.38	19.30 ± 1.30	21.36 ± 0.48	20.72 ± 0.87	22.64 ± 0.34	22.18 ± 0.80
	*p* =0.19		*p* = 0.92		*p* = 0.29		*p* = 0.13		*p* = 0.34	
**ITD** (mm)	24.82 ± 0.30	24.13 ± 1.32	28.94 ± 0.70	27.65 ± 1.61	31.16 ± 0.38	29.63 ± 1.20	32.66 ± 0.92	30.35 ± 1.50	32.98 ± 0.32	31.95 ± 1.21
	*p* = 0.28		*p* = 0.12		*p* = 0.19		*p* = 0.01		*p* = 0.09	

CG—control group, EG—experimental group, AD—axial diameter, SSD—superior sagittal diameter, SD4—sagittal diameter at the level of 4th rib, ISD—inferior sagittal diameter, STD—superior transverse diameter, TD4—transverse diameter at the level of 4th rib, ITD—inferior transverse diameter.

**Table 2 life-10-00096-t002:** Haller index preoperative (CT1) through 4 weeks after surgery.

	CT1 (Preoperative)		CT2 (7 Days Postop)		CT3 (14 Days Postop)		CT4 (21 Days Postop)		CT5 (28 Days Postop)	
	CG	EG	CG	EG	CG	EG	CG	EG	CG	EG
**SHI**	1.45 ± 0.02	1.38 ± 0.03	1.40 ± 0.08	1.44 ± 0.12	1.31 ± 0.08	1.38 ± 0.17	1.37 ± 0.07	1.35 ± 0.07	1.42 ± 0.09	1.26 ± 0.11
	*p* = 0.03		*p* = 0.49		*p* = 0.38		*p* = 0.68		*p* = 0.02	
**HI4**	1.75 ± 0.07	1.61 ± 0.13	1.65 ± 0.07	1.61 ± 0.14	1.65 ± 0.07	1.74 ± 0.14	1.68 ± 0.07	1.63 ± 0.12	1.62 ± 0.03	1.61 ± 0.08
	*p* = 0.22		*p* = 0.52		*p* = 0.16		*p* = 0.41		*p* = 0.70	
**IHI**	1.43 ± 0.03	1.50 ± 0.10	1.18 ± 0.04	1.49 ± 0.23	1.10 ± 0.10	1.49 ± 0.23	1.08 ± 0.04	1.46 ± 0.15	1.16 ± 0.10	1.41 ± 0.11
	*p* = 0.20		*p* = 0.03		*p* = 0.00		*p* = 0.00		*p* = 0.04	

CG—control group, EG—experimental group, SHI—Haller index at the superior thoracic aperture, HI4—Haller index at the level of 4th rib, IHI—Haller index at the inferior thoracic aperture.

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
