# Peer review of "Rat Animal Model of Pectus Excavatum"

_life, 2020, doi:10.3390/life10060096_

Round 1
Reviewer 1 Report
Background: Pectus Excavatum (PE) is the most common congenital deformity of the thoracic wall. Lately significant achievements have been made in finding new, less invasive treatment methods for PE. However most of the experimental work has been carried out without the help of an animal model. This report describes a method to create an animal model for PE in Sprague Dowley rats.
Methods: 15 Sprague Dowley rat pups were divided into two groups: 10 for the experimental group (EG) and 5 for the control group (CG). The last 4 pairs of costal cartilages in rats in the EG were resected. The animals were assessed by CT-scan prior to surgery and weekly for 4 consecutive weeks. After 4 weeks the animals were euthanized and the thoracic cage was dissected from the surrounding tissue.
Results: On the first postoperative CT, 7 days after surgery we observed a marked depression of the lower sternum in all animals from EG. This deformity was present at every CT-scan after surgery and at the post-euthanasia assessment.
Conclusions: By decreasing the structural resistance of the lower costal cartilages we produced a PE animal model in Sprague-Dowley rats.
General Comments:
Interesting study. Well-designed and well-written. Needs minor editing of English.
Specific comments:
1. The authors state “Interventionary studies involving animals or humans and other studies require ethical approval must list the authority that provided approval and the corresponding ethical approval code.” Was approval obtained from their institutional review board or ethics committee or not? One cannot tell from this statement. There should also be approval from the animal experimentation committee which is a separate authority from the IRB.
2. PE has clinical implications. The main reason why thoracic dimensions change is because of alterations in respiratory mechanics. Individuals with PE are vulnerable to respiratory complications including dyspnea, respiratory failure and pneumonia. PE results in reduction in thoracic compliance (and maybe resistance). Therefore, it would have been useful to measure respiratory mechanics during anesthesia by recording airway pressure, volume and flow. Hopefully this data is still available from the time the rats were anesthetized and operated on. From these values, respiratory compliance and airflow resistance can be computed. There are many papers in the literature describing how such measurements are made, including in other chest wall deformities such as scoliosis. Check papers by Bergofsky (Medicine, 1959), Baydur et al (Journal of Applied Physiology, 1980s and 1990s). Findings also have implications for clinical medicine and orthopedics.
3. The authors can then compare any changes in the respiratory mechanics following the resections of the costal cartilages. One would expect that respiratory compliance increased and resistance possibly also decreased. If the data are available, the authors may have serial measurements of these variable over the time the rats remained under anesthesia. Then changes in the mechanics may correlate with changes in the dimensions of the rib cage.
4. Another important feature the authors could record is how much anterior displacement of the lower rib cage there was post-surgery while the animals were breathing spontaneously. Did the rats exhibit “paradoxical” breathing during inspiratory efforts more than before the surgery?
5. The authors use the wrong term of “resistance” for chest wall structures. Resistance is defined as airway pressure/flow. The correct term to use is elastance or stiffness (or its reciprocal value, compliance).
6. The authors should describe how these changes in chest wall mechanics may be applicable to clinical situations.
7. “Tranverse” is spelled with an “e” at the end.
8. “Interventionary studies…” should be changed to “interventional…”.
Author Response
- The authors state “Interventionary studies involving animals or humans and other studies require ethical approval must list the authority that provided approval and the corresponding ethical approval code.” Was approval obtained from their institutional review board or ethics committee or not? One cannot tell from this statement. There should also be approval from the animal experimentation committee which is a separate authority from the IRB.
The project was approved by the Ethics Committee of the “Victor Babes” University of Medicine and Pharmacy Timisoara. The reference number was added to the text
- PE has clinical implications. The main reason why thoracic dimensions change is because of alterations in respiratory mechanics. Individuals with PE are vulnerable to respiratory complications including dyspnea, respiratory failure and pneumonia. PE results in reduction in thoracic compliance (and maybe resistance). Therefore, it would have been useful to measure respiratory mechanics during anesthesia by recording airway pressure, volume and flow. Hopefully this data is still available from the time the rats were anesthetized and operated on. From these values, respiratory compliance and airflow resistance can be computed. There are many papers in the literature describing how such measurements are made, including in other chest wall deformities such as scoliosis. Check papers by Bergofsky (Medicine, 1959), Baydur et al (Journal of Applied Physiology, 1980s and 1990s). Findings also have implications for clinical medicine and orthopedics.
We agree that respiratory mechanics and the alterations of the respiratory mechanism is an important aspect in the pathogeny of PE. Unfortunately, these data are not available. As the purpose of the study was to design an animal model for PE we did not record these data.
- The authors can then compare any changes in the respiratory mechanics following the resections of the costal cartilages. One would expect that respiratory compliance increased and resistance possibly also decreased. If the data are available, the authors may have serial measurements of these variable over the time the rats remained under anesthesia. Then changes in the mechanics may correlate with changes in the dimensions of the rib cage.
AS with previous point, we did not recorded and analyzed these data. Our focus was on obtaining a valid animal model.
- Another important feature the authors could record is how much anterior displacement of the lower rib cage there was post-surgery while the animals were breathing spontaneously. Did the rats exhibit “paradoxical” breathing during inspiratory efforts more than before the surgery?
We did not noticed instant paradoxical breathing movements after surgical procedure.
- The authors use the wrong term of “resistance” for chest wall structures. Resistance is defined as airway pressure/flow. The correct term to use is elastance or stiffness (or its reciprocal value, compliance).
We would prefer to use the term “structural strength”, because it refers to the ability of the costal cartilages to sustain the normal shape of the thorax.
- The authors should describe how these changes in chest wall mechanics may be applicable to clinical situations.
In the first paragraph of the discussion section we described the clinical situation when there is an imbalance between the inner thoracic inspiratory forces and the structural strength of the anterior chest wall. How the mechanical weak costal cartilages may be displaced posteriorly by this forces during inspiration.
- “Tranverse” is spelled with an “e” at the end.
We have made the correction
- “Interventionary studies…” should be changed to “interventional…”.
We have made the correction
Reviewer 2 Report
This study creates an animal model of pectus excavatum (PE) in rats. The authors resected the last 4 costal cartilages hypothesizing that it would lead to PE.
Comments:
- It is not clearly explained what the resection exactly simulates. My understanding is that the resection created mechanically weakened structure in the lower area of the sternum. It seems to be influenced by the study performed by Feng et al. However, the tissue in that study was frozen, stored, and then thawed which can lead to changes in mechanical properties that were reported.
- PE in the majority of cases is asymmetric and characterized by the sternum has torsion. That supports the argument of overgrown costal cartilage. Could you address how your methodology is dealing with this hypothesis?
- Typos, grammar, and language issues
- Section 2.2. and others: it is advised to use the past tense when reporting the procedure that was followed. There are inconsistencies in tenses used.
- line 42: is: associate, should be: associated
- line 52: 'unknown' rather than 'an unresolved issue' could be a better choice and less wordy.
- line 51-52: commas missing after 'animals' and 'dogs'
- Section 2.3 should be on the next page
- Line 79: comma missing after 'scan'
- Line 81: space missing after 'SD4)'
- Line 97: Space missing before 'Interventionary'
- Section 3: please revise spaces before the number and before and after '+/-'
- Line 108: missing comma before 'we'
- Line 116: suggested replacement of 'all through' with 'throughout'. If 'all through' is kept, it should be hyphenated.
- Lines 116-122: spaces before and after '>' signs
- Table 1: check spaces between '7' and 'days'
- Line 131: inconsistent spacing between the number and the '>' sign.
- Table 2: check spaces between '7' and 'days'
- Lines 143-146: inconsistent spacing
- Line 145: consider changing 'the difference being' to 'which was'
- Line 151: suggestion: change 'diseases' to 'medical conditions'
- Line 153: check spacing before 'Our'
- Line 158: too many periods
- Line 166-167: missing commas.
Author Response
- It is not clearly explained what the resection exactly simulates. My understanding is that the resection created mechanically weakened structure in the lower area of the sternum. It seems to be influenced by the study performed by Feng et al. However, the tissue in that study was frozen, stored, and then thawed which can lead to changes in mechanical properties that were reported.
The experiment is based on the assumption that a weak (weakened) costal cartilages induce the collapse of the anterior chest wall due to intrathoracic negative pressure. We added an explanation in the discussion section, paragraph 2.
- PE in the majority of cases is asymmetric and characterized by the sternum has torsion. That supports the argument of overgrown costal cartilage. Could you address how your methodology is dealing with this hypothesis?
This is a very interesting issue. In our opinion, the asymmetric displacement of the sternum may happen in this instance as well because, in in vivo situation the costal cartilages does not necessary have to exhibit similar grades of “weakness”. Also, the forces acting over the chest wall may not be even distributed over the thoracic wall. Unfortunately, our methodology does not mimic precisely the natural pathogenic process of PE in humans. For that we have to find a method to induce a “less sudden”, more controlled, reduction of the structural strength of the costal cartilages. WE added a comment on this matter when we discussed the limitations of our study.
Section 2.2. and others: it is advised to use the past tense when reporting the procedure that was followed. There are inconsistencies in tenses used.
We changed all sentences to past perfect
line 42: is: associate, should be: associated
Corrected
line 52: 'unknown' rather than 'an unresolved issue' could be a better choice and less wordy
Corrected
line 51-52: commas missing after 'animals' and 'dogs'
Corrected
Section 2.3 should be on the next page
Corrected
Line 79: comma missing after 'scan'
Corrected
Line 81: space missing after 'SD4)'
Corrected
Line 97: Space missing before 'Interventionary'
Corrected
Section 3: please revise spaces before the number and before and after '+/-'
Corrected
Line 108: missing comma before 'we'
Corrected
Line 116: suggested replacement of 'all through' with 'throughout'. If 'all through' is kept, it should be hyphenated.
Corrected
Lines 116-122: spaces before and after '>' signs
Corrected
Table 1: check spaces between '7' and 'days'
Corrected
Line 131: inconsistent spacing between the number and the '>' sign.
Corrected
Table 2: check spaces between '7' and 'days'
Corrected
Lines 143-146: inconsistent spacing
Corrected
Line 145: consider changing 'the difference being' to 'which was'
Corrected
Line 151: suggestion: change 'diseases' to 'medical conditions'
Corrected
Line 153: check spacing before 'Our'
Corrected
Line 158: too many periods
Corrected
Line 166-167: missing commas.
Corrected
Reviewer 3 Report
line 42: associated
line 67: what effect might the detachment of the pectoralis major muscle have had on the results?
line 92: HIS should be SHI as used elsewhere in the paper
line 93: HII should be IHI
line 117: should be "from" instead of "form"
line 120: Fig.2 should be Fig. 3
line 131: "groups" instead of "group"
Figure 4: Vertical axis should be IHI values
line 163: Brown reference missing or incorrect.
line 180: "these" instead of "this"
line 182: "conditions" instead of "condition"
Author Response
line 42: associated
Corrected
line 67: what effect might the detachment of the pectoralis major muscle have had on the results?
We are not able to precisely quantify the effect of muscle detachment over the results. We stated that this might be a limitation of the study: that local effects of the surgical procedures may interfere with the whole process. However, the surgical dissection was kept to the minimum and re-attached the pectoral muscle after surgery. So, we believe that from the mechanical point of view, the detachment of the muscle had no influenced. However, the local scaring process might have. We added explanations in text.
line 92: HIS should be SHI as used elsewhere in the paper
Corrected
line 93: HII should be IHI
Corrected
line 117: should be "from" instead of "form"
Corrected
line 120: Fig.2 should be Fig. 3
Corrected
line 131: "groups" instead of "group"
Corrected
Figure 4: Vertical axis should be IHI values
Corrected
line 163: Brown reference missing or incorrect.
Corrected
line 180: "these" instead of "this"
Corrected
line 182: "conditions" instead of "condition"
Corrected
Round 2
Reviewer 1 Report
The authors have responded to the reviewer's comments in satisfactory manner.